# Affinity and Specificity for Binding to Glycosaminoglycans Can Be Tuned by Adapting Peptide Length and Sequence

**DOI:** 10.3390/ijms23010447

**Published:** 2021-12-31

**Authors:** Helena Crijns, Lowie Adyns, Eva Ganseman, Seppe Cambier, Eline Vandekerckhove, Noëmie Pörtner, Lotte Vanbrabant, Sofie Struyf, Tanja Gerlza, Andreas Kungl, Paul Proost

**Affiliations:** 1Laboratory of Molecular Immunology, Department of Microbiology, Immunology and Transplantation, Rega Institute for Medical Research, KU Leuven, 3000 Leuven, Belgium; helena.crijns@kuleuven.be (H.C.); lowie.adyns@kuleuven.be (L.A.); eva.ganseman@kuleuven.be (E.G.); seppe.cambier@kuleuven.be (S.C.); eline.vandekerckhove@edu.vlerick.com (E.V.); noemie.portner@kuleuven.be (N.P.); lotte.vanbrabant@kuleuven.be (L.V.); sofie.struyf@kuleuven.be (S.S.); 2Department of Pharmaceutical Chemistry, Institute of Pharmaceutical Sciences, Karl-Franzens Universität, 8010 Graz, Austria; tanja.gerlza@uni-graz.at (T.G.); andreas.kungl@uni-graz.at (A.K.)

**Keywords:** heparan sulfate, heparin, hyaluronic acid, glycosaminoglycan, peptides, glycosaminoglycan-binding motif, chemokine

## Abstract

Although glycosaminoglycan (GAG)–protein interactions are important in many physiological and pathological processes, the structural requirements for binding are poorly defined. Starting with GAG-binding peptide CXCL9(74-103), peptides were designed to elucidate the contribution to the GAG-binding affinity of different: (1) GAG-binding motifs (i.e., BBXB and BBBXXB); (2) amino acids in GAG-binding motifs and linker sequences; and (3) numbers of GAG-binding motifs. The affinity of eight chemically synthesized peptides for various GAGs was determined by isothermal fluorescence titration (IFT). Moreover, the binding of peptides to cellular GAGs on Chinese hamster ovary (CHO) cells was assessed using flow cytometry with and without soluble GAGs. The repetition of GAG-binding motifs in the peptides contributed to a higher affinity for heparan sulfate (HS) in the IFT measurements. Furthermore, the presence of Gln residues in both GAG-binding motifs and linker sequences increased the affinity of trimer peptides for low-molecular-weight heparin (LMWH), partially desulfated (ds)LMWH and HS, but not for hyaluronic acid. In addition, the peptides bound to cellular GAGs with differential affinity, and the addition of soluble HS or heparin reduced the binding of CXCL9(74-103) to cellular GAGs. These results indicate that the affinity and specificity of peptides for GAGs can be tuned by adapting their amino acid sequence and their number of GAG-binding motifs.

## 1. Introduction

Glycosaminoglycans (GAGs) are large, complex carbohydrates with molecular weights ranging from approximately 10 to 100 kDa. In short, GAGs are linear, negatively charged polysaccharides composed of repeating disaccharide units. Six different types of GAGs can be distinguished: heparin, heparan sulfate (HS), chondroitin sulfate (CS), dermatan sulfate (DS), keratan sulfate (KS) and hyaluronic acid (HA). All GAGs, with the exception of HA, are sulfated at various positions and are covalently attached to core proteins, thereby forming proteoglycans [1]. Proteoglycans are produced by virtually all mammalian cells and can be stored in secretory granules, inserted into the plasma membrane or secreted into the extracellular matrix (ECM) [2].

Recently, Vallet et al. described the first draft of the GAG interactome, which comprises 827 proteins and 932 GAG–protein interactions [3]. Indeed, GAGs can bind to a variety of different proteins, such as growth factors, cytokines, chemokines, enzymes and adhesion molecules. These interactions enable them to play a major role in various physiological processes; for instance, embryonic development, assembly of the ECM, cell signaling, cell proliferation, angiogenesis and anti-coagulation. In addition, GAG–protein interactions are involved in many pathological contexts and human diseases, such as tumor progression, metastasis, inflammation, infectious diseases, mucopolysaccharidoses, and cardiovascular and neurodegenerative diseases. Hence, targeting specific GAG–protein interactions may be a valuable strategy for use in therapeutic drug development [1,3,4].

In general, the binding of GAGs with proteins is considered to be based mainly on electrostatic interactions between negatively charged sulfate or carboxylate groups of GAGs and positively charged amino acids in proteins. These amino acids include Arg, Lys and, to a lesser extent, His. Besides electrostatic interactions, van der Waals forces, hydrogen bonds and hydrophobic interactions with the carbohydrate backbone are also involved in the binding of proteins to GAGs [1,5,6].

Several amino acid consensus sequences have been demonstrated or have been predicted to be the sequences in proteins that interact with GAGs. As positively charged amino acids are clearly of major importance in GAG binding, these consensus sequences are characterized by specific arrangements of basic amino acids. Cardin and Weintraub investigated the amino acid sequences of 21 heparin-binding proteins and determined two consensus sequences for the recognition of GAGs: XBBXBX and XBBBXXBX [7]. B and X represent basic and hydropathic amino acid residues, respectively. The basic residues were most commonly Lys or Arg, as opposed to His, which was infrequently found. The amino acid residues Asn, Ser, Ala, Gly, Ile, Leu and Tyr were relatively abundant at the X positions in the XBBBXXBX consensus sequence, whereas Cys, Glu, Asp, Met, Phe and Trp rarely occurred at these positions [7]. Sobel et al. assessed the amino acid sequences of the established heparin-binding domains of several proteins and observed a clear pattern of clustered cationic residues that are separated by predominantly non-polar, neutral amino acid residues. As a result, they proposed an optimal, palindromic consensus sequence: XBBXXBBBXXBBX, where cationic residues are represented by B. Moreover, they identified a 23-residue heparin-binding domain in human von Willebrand factor (Tyr565–Ala587), which closely approximates the proposed consensus sequence [8]. Furthermore, a structural analysis of the acidic fibroblast growth factor (FGF), basic FGF and transforming growth factor-β1 (TGF-β1) by Hileman et al. led to the description of another type of consensus sequence: TXXBXXTBXXXTBB. In this consensus sequence, turns, basic amino acid residues (Arg, Lys or occasionally Gln) and hydropathic amino acid residues are denoted by T, B and X, respectively. Basic amino acid residues are brought into proximity by the turns, which emphasizes the importance of the secondary structure in GAG-binding motifs [9]. In addition, Torrent et al. searched for a conserved structural pattern in 20 heparin-binding proteins and reported a cation-polar-cation (CPC) clip motif in heparin-binding proteins. This structural motif consists of one polar (preferentially Asn, Gln, Thr, Tyr or Ser) and two positively charged amino acid residues (Arg or Lys). Moreover, the spatial arrangement of this CPC clip motif is characterized by regular distances between the α carbons as well as the center of gravity of the side chains of the amino acid residues. This structural motif acts as a conserved structural signature for heparin-binding proteins [10]. Rudd et al. sought potentially conserved HS/heparin-binding motifs in 437 HS/heparin-binding proteins and concluded that these motifs may be formed on protein surfaces by short, spaced, basic amino acid sequences in a three-dimensional arrangement [11]. This study, as well as examples such as antithrombin and XCL1 (lymphotactin), indicates that not only the primary sequence, but also the tertiary structure of proteins determines GAG–protein interactions [12,13,14]. The spatial proximity of the basic residues in the primary amino acid sequence is therefore not essential.

Nevertheless, despite research concerning the molecular mechanisms underlying GAG–protein interactions, the structural requirements of proteins for binding to GAGs are still not well defined. In view of this, detailed studies of the elements contributing to GAG binding are valuable. Therefore, in this study, we aimed to elucidate the structural requirements for the binding of proteins to GAGs. Considering this, we investigated the role of consensus sequences, hereafter referred to as “GAG-binding motifs”, in the interactions between GAGs and peptides. The chemokine-derived peptide CXCL9(74-103) was previously synthesized in our laboratory and consists of the 30 COOH-terminal amino acids of the chemokine CXCL9. Remarkably, more than 50% of the amino acids of this peptide are positively charged and it contains two GAG-binding motifs: ^75^KKQK^78^ (BBXB) and ^85^KKKVLK^90^ (BBBXXB). Several studies have demonstrated the binding of this peptide to GAGs with high affinity. Moreover, this CXCL9-derived peptide could compete with several chemokines and vascular endothelial growth factor (VEGF) for binding to GAGs [15,16,17,18,19,20]. Therefore, CXCL9(74-103) served as a starting point for our research concerning the elements contributing to GAG binding. In essence, different peptides with varying lengths and varying numbers and types of GAG-binding motifs were designed. In order to evaluate the contribution of specific amino acids to GAG binding, we altered the amino acid sequences of the GAG-binding motifs and the sequences in between the GAG-binding motifs, hereafter referred to as “linker sequences”. The peptides were chemically synthesized and their binding affinity for various GAGs (low-molecular-weight heparin (LMWH), partially desulfated LMWH (dsLMWH), HS and HA) was determined using isothermal fluorescence titration (IFT). In addition, we assessed the binding of several of these peptides to cellular GAGs on Chinese hamster ovary (CHO) cells with flow cytometry. Moreover, the binding of the peptides to cellular GAGs in the presence of soluble GAGs was investigated.

## 2. Results

### 2.1. Chemical Synthesis and Fluorescent Labeling of Peptides

The peptide CXCL9(74-103) has been previously studied and has been shown to bind to GAGs with high affinity [15,16,17,18,19]. Based on this knowledge, we wanted to investigate the requirements for high affinity binding to GAGs, more specifically: if and how (1) the non-basic amino acids in the GAG-binding motifs; (2) the amino acids in the linker sequences; (3) the number of GAG-binding motifs; and (4) the type of GAG-binding motifs affect the affinity of binding to various GAGs. To this end, seven peptides with varying lengths and varying numbers and types of GAG-binding motifs and hydrophilic (Gln and Ser) or hydrophobic (Ala) amino acids in the linker sequences and GAG-binding motifs were designed. Peptides with two or three GAG-binding motifs were called dimers (2mers) or trimers (3mers), respectively. Either ASAS or QSQS was used as linker sequences, where A, Q and S represent the amino acids Ala, Gln and Ser, respectively. In accordance with CXCL9(74-103), the COOH-terminus of all the peptides ended with two Thr residues. These peptides were chemically synthesized and site-specifically labeled with a fluorescent group, 5(6)-carboxytetramethylrhodamine (TAMRA), at the NH_2_-terminus. After purification by RP-HPLC, the relative molecular mass (M_r_) of each peptide was confirmed by ion trap mass spectrometry. The differences between the theoretical and experimentally determined M_r_ were within the error margins of the mass spectrometer. For each peptide, the amino acid sequence, the theoretical and the experimentally determined M_r_ are shown in Table 1. A mass spectrum of one of the peptides, TAMRA-KKQK-3mer, is depicted in Figure 1.

In summary, these chemically synthesized peptides with: (1) different amino acids in the GAG-binding motifs and alternative amino acids in the linker sequences (i.e., Ala or Gln); (2) a different number of GAG-binding motifs and, consequently, different lengths (i.e., dimers and trimers); and (3) different GAG-binding motifs (i.e., short and long GAG-binding motifs) allowed us to investigate in detail the contribution of amino acids in GAG-binding motifs and linker sequences to the affinity for GAGs.

### 2.2. An Increased Number of GAG-Binding Motifs Results in Higher Affinity for HS

To assess the affinity of the peptides for different GAGs, IFT measurements were performed with HS, LMWH, dsLMWH and HA. The fluorescence of the conjugated fluorophore TAMRA at the NH_2_-terminus of the peptides was detected.

First, IFT measurements were performed to evaluate the binding of the different TAMRA-labeled peptides to HS. The binding isotherms and their corresponding dissociation constant (K_d_) values are shown in Figure 2A,B, respectively. The binding isotherms of the trimer peptides (KKAK-, KKQKASAS- and KKQK-3mer) are characterized by steeper slopes in comparison with the binding isotherms of the dimer peptides (KKAK-, KKQKASAS-, KKQK- and KKKQQK-2mer). Accordingly, the K_d_ values of the trimer peptides are lower than those of the dimer peptides, indicating higher affinity for HS. Furthermore, CXCL9(74-103) showed the highest affinity for HS. In view of this, peptide length and the number and type of GAG-binding motifs seem to contribute to HS binding affinity, as longer peptides with extra GAG-binding motifs tend to show higher affinity for HS. 

Additional IFT measurements were performed with LMWH, dsLMWH and HA for all TAMRA-labeled peptides. The binding isotherms and K_d_ values for LMWH are depicted in Figure 2C,D. In comparison with HS, the difference between dimer and trimer peptides is less pronounced on LMWH, especially for the KKAK-peptides. The affinity of the KKQK-3mer is similar for HS and LMWH. Furthermore, the peptide with the highest affinity for LMWH is CXCL9(74-103). In addition, with dsLMWH (Figure 2E,F), the highest affinity was measured for CXCL9(74-103). Remarkably, the peptide with the highest affinity for HA (Figure 2G,H) appeared to be the KKAK-3mer and not CXCL9(74-103).

### 2.3. Gln Residues in Both GAG-Binding Motifs and Linker Sequences Increase the Affinity of Trimer Peptides for LMWH, dsLMWH and HS, but Not for HA

The IFT measurements of the three trimer peptides (KKAK-, KKQK- and KKQKASAS-3mer) allowed us to evaluate the contribution of different amino acids to GAG-binding affinity. The binding isotherms and their corresponding K_d_ values are shown in Figure 3A–C and Figure 3D, respectively. The effect of the presence of hydrophobic (Ala) versus hydrophilic (Gln) residues in the GAG-binding motifs could be investigated by comparing the KKAK-3mer (Figure 3A) with the KKQKASAS-3mer (Figure 3B). The replacement of Ala by Gln residues in the GAG-binding motifs, generating the KKQKASAS-3mer, resulted in increased affinity of the peptide for LMWH, dsLMWH and HS. In contrast, a decreased affinity of the KKQKASAS-3mer for HA was observed. Furthermore, the KKQKASAS- and KKQK-3mers were compared to evaluate the contribution of Ala and Gln residues in the linker sequences to GAG-binding affinity. The KKQK-3mer (Figure 3C) showed higher affinity for LMWH, dsLMWH and HS, whereas the KKQKASAS-3mer showed higher affinity for HA. In conclusion, amino acids in both GAG-binding motifs and linker sequences appear to affect GAG-binding affinity. Moreover, hydrophilic Gln residues in GAG-binding motifs and linker sequences seem to increase the binding affinity for LMWH, dsLMWH and HS. However, the presence of hydrophobic Ala residues in GAG-binding motifs and linker sequences improves the affinity for HA.

### 2.4. Dimer Peptides with Long versus Short GAG-Binding Motifs Are Characterized by Differential Affinity for LMWH and dsLMWH

The contribution of short versus long GAG-binding motifs to GAG-binding affinity was investigated by comparing the KKQK-2mer and the KKKQQK-2mer, containing two short and two long GAG-binding motifs, respectively. Figure 4A,B represent the binding isotherms of these peptides, and their K_d_ values are displayed in Figure 4C. The KKKQQK-2mer was characterized by a higher affinity for LMWH and dsLMWH in comparison with the KKQK-2mer, indicating a possible role of long GAG-binding motifs for binding to these GAGs. For HS and HA, only minor differences between the two peptides were observed.

A summarizing overview of all the K_d_ values, derived from the IFT measurements, is depicted in Figure 5. This comparison shows that the chemokine-derived CXCL9(74-103) has the highest affinity for GAGs and that, in contrast to almost all other peptides, CXCL9(74-103) has a high preference for binding to HS compared with other GAGs.

### 2.5. Subcloning of a Mixed Population of pgsB-618 Cells Results in HS^+^ and HS^−^ Subclones

The IFT experiments already provided insight into the interaction between the peptides and the GAGs. Subsequently, we wanted to investigate this interaction in a physiologically more relevant setting, i.e., the binding of the peptides to cellular GAGs. Therefore, we used CHO-K1 cells and pgsB-618 cells in parallel for the experiments. According to Expasy [21], pgsB-618 cells are mutant CHO-K1 cells that are deficient in galactosyltransferase I, also known as β-1,4-galactosyltransferase 7 (β4Gal-T7). This enzyme is required for the synthesis of the common tetrasaccharide linkage region in the biosynthesis of proteoglycans [22,23]. Consequently, pgsB-618 cells are proteoglycan-deficient mutants. Upon receiving the pgsB-618 cells, their HS expression was evaluated using flow cytometry to confirm their inability to produce GAGs. Remarkably, 36.8% of the pgsB-618 cells stained positive for HS on their cell surface using flow cytometry, suggesting a mixed population of cells, consisting of both HS^+^ and HS^−^ subpopulations (Appendix A). Two weeks later, the HS^+^ subpopulation of the pgsB-618 cells even increased to 91.9% (Appendix A). The expression of HS on CHO-K1 cells was also measured and, as expected, a uniform HS^+^ cell population was observed (Appendix A). PgsB-618 cells, deficient in GAG biosynthesis, were required for the experiments to serve as an appropriate negative control. To this end, subcloning of the pgsB-618 cells was performed and resulted in 21 subclones, each derived from a single cell. The HS expression on each of the 21 pgsB-618 subclones was evaluated using flow cytometry immediately after subcloning and appeared to be absent in 7 out of 21 subclones. One HS^−^ subclone (clone 16) was selected for use in further experiments. In addition, one HS^+^ subclone (clone 19) was chosen (Appendix A). More than 3 months and 36 cell passages later, the HS expression was measured again, and the stability of both cell lines was confirmed (Figure 6). For reasons of simplicity, HS^−^ pgsB-618 clone 16 cells and HS^+^ pgsB-618 clone 19 cells will be named HS^−^ and HS^+^ CHO cells, respectively, in the following paragraphs.

### 2.6. Peptides Bind to Cellular GAGs with Differential Capacity

Based on the IFT data, four peptides were selected to investigate their binding to cellular GAGs on CHO cells. A fixed dose (0.24 nmol) of TAMRA-labeled peptide (KKAK-2mer, KKQK-3mer, KKKQQK-2mer or CXCL9(74-103)) was added to the cells. Histograms resulting from the flow cytometry analysis are shown in Figure 7A,B for HS^+^ and HS^−^ CHO cells, respectively. The fluorescence intensity on the HS^−^ CHO cells is shifted to the left compared with the HS^+^ CHO cells, indicating a decreased binding of the TAMRA-labeled peptides to the HS^−^ CHO cells. Remarkably, the binding of the TAMRA-labeled peptides to the HS^−^ CHO cells was not completely abolished in comparison with the HS^+^ CHO cells. The median fluorescence intensity (MFI) values were calculated by subtracting the MFI of the unstained sample from the MFI of the sample with TAMRA-labeled peptide, per cell type (HS^+^ or HS^−^ CHO cells) and compared for the different TAMRA-labeled peptides. All four peptides that are shown in Figure 7 bound significantly better to HS^+^ CHO cells in comparison with HS^−^ CHO cells (data not shown; Wilcoxon test; KKAK-2mer, *p* < 0.05; KKQK-3mer, *p* < 0.001; KKKQQK-2mer, *p* < 0.0001; CXCL9(74-103), *p* < 0.0001). In addition, in order to compare the different peptides, the MFI values of the TAMRA-labeled peptide bound to the HS^−^ CHO cells were subtracted from those of the HS^+^ CHO cells, per peptide. The resulting values (MFI ([HS^+^]–[HS^−^])) for each peptide are shown in Figure 7C. The binding capacity of CXCL9(74-103) for cellular GAGs was significantly higher (*p* < 0.0001) compared with that of the three other peptides (KKAK-2mer, KKQK-3mer and KKKQQK-2mer). Furthermore, the capacities of both the KKKQQK-2mer and the KKQK-3mer for binding to cellular GAGs were significantly higher (*p* < 0.05) in comparison with the binding capacity of the KKAK-2mer. To summarize, the order of increasing capacity of the TAMRA-labeled peptides for binding to cellular GAGs is: KKAK-2mer < KKQK-3mer < KKKQQK-2mer < CXCL9(74-103).

### 2.7. Binding of CXCL9(74-103) to Cellular GAGs Is Reduced by Adding Heparin or HS

In the next step, we studied the effect of adding soluble GAGs in combination with TAMRA-labeled peptides to the CHO cells. To this end, one GAG (heparin, HS or HA) and one peptide (KKQK-3mer, KKKQQK-2mer or CXCL9(74-103)) were simultaneously added to either HS^+^ or HS^−^ CHO cells. Different doses of each GAG (0, 0.03, 0.1, 0.3, 1 and 3 μg) and one dose of the peptide (0.24 nmol) were used. The MFI values of the different TAMRA-labeled peptides that are bound to the HS^+^ or HS^−^ CHO cells are shown in Appendix A. The net MFI values were calculated by subtracting the MFI of the unstained sample from the MFI of the sample with TAMRA-labeled peptides, per cell type (HS^+^ or HS^−^ CHO cells). First, the binding of each peptide to either the HS^+^ or HS^−^ CHO cells was compared per dose of GAG. Although the absolute MFI values were not largely different for both the KKQK-3mer and the KKKQQK-2mer, a significantly higher amount of each peptide was bound to HS^+^ CHO cells compared with HS^−^ CHO cells (Appendix A). Second, the different doses of GAG were compared with the condition without GAG. This comparison was performed per GAG and per peptide. The MFI values of the TAMRA-labeled peptides bound to the HS^−^ CHO cells were subtracted from those of the HS^+^ CHO cells, per peptide and per (dose of) GAG. The resulting values (MFI ([HS^+^]–[HS^−^]); data not shown) were used for statistical analysis. A Kruskal–Wallis test showed no significant differences between the groups for the KKQK-3mer (heparin, HS and HA; Appendix A), the KKKQQK-2mer (heparin, HS and HA; Appendix A) and CXCL9(74-103) (HA; Appendix A). Hence, in these cases, the addition of soluble GAGs failed to reduce the binding of the peptides to the CHO cells. In contrast, a statistically significant reduction of binding was detected for CXCL9(74-103) with heparin (Appendix A) and with HS (Appendix A) (both *p* < 0.0001). Subsequently, a Mann–Whitney test was applied to compare each individual dose of heparin and HS to the condition without GAG for CXCL9(74-103) (Appendix A). Interestingly, the addition of heparin (the three highest doses: 0.3, 1 and 3 μg) significantly reduced the binding of CXCL9(74-103) to the CHO cells (Appendix A). A similar result was observed for HS (Appendix A), although only the two highest doses (1 and 3 μg) could significantly reduce the binding of the peptide to the cellular GAGs. As opposed to heparin and HS, the addition of HA did not affect the binding of CXCL9(74-103) to the cellular GAGs on the CHO cells (Appendix A).

## 3. Discussion

Chemokines are small proteins that have two main interaction partners: chemokine receptors, which are G protein-coupled receptors (GPCRs), and GAGs [24,25]. The binding of chemokines to GAGs enables the formation of a chemokine concentration gradient, which is indispensable for leukocyte migration [26]. Interfering with the interactions between GAGs and chemokines can be a useful therapeutic strategy to inhibit inflammation [27]. The chemokine CXCL9 binds to CXCR3 and is characterized by a highly positively charged COOH-terminal tail. A CXCL9-derived peptide, CXCL9(74-103), consisting of the COOH-terminal region of CXCL9 has been synthesized previously in our laboratory. This peptide lacks the capacity to bind to, and signal through, CXCR3 [15]. Interestingly, high affinity of CXCL9(74-103) for GAGs has already been shown in previous studies [15,16,17,18,19]. Since we aimed to elucidate the structural requirements of proteins for binding to GAGs, we started our research from this known GAG-binding peptide. As CXCL9(74-103) contains the two different GAG-binding motifs described by Cardin and Weintraub: BBXB and BBBXXB, we selected these two motifs to be part of the amino acid sequences of seven new peptides (Table 1) [7]. Fromm et al. investigated the interactions of Arg and Lys with heparin [28]. A higher affinity of Arg-containing peptides for heparin was observed in comparison with Lys-containing peptides. In addition, acetyl-Arg-amide bound approximately 2.5 times more tightly to heparin than the corresponding acetyl-Lys-amide. This study suggests that the difference between these amino acids is due to the intrinsic properties of their side chains. To be specific, the tighter interaction between the guanidino group of the Arg side chain and the sulfates in heparin may be due to stronger hydrogen bonding and a more exothermic electrostatic interaction [28]. Despite this knowledge, we designed peptides with Lys and not Arg residues as basic amino acids because CXCL9(74-103) contains mainly Lys residues (13 Lys versus 3 Arg residues). Moreover, the GAG-binding motifs that are present in this CXCL9-derived peptide only contain Lys residues as basic amino acids.

In order to study the interactions between several GAGs ((ds)LMWH, HS and HA) and peptides, IFT experiments were performed, and the resulting binding isotherms were used to calculate the K_d_ values. In general, LMWH is more sulfated than HS, which consists of sulfated, non-sulfated and mixed domains [1,29]. The dsLMWH is partially (2-O, 3-O) desulfated LMWH, but is expected to be still more sulfated than HS, although the sulfated domains of HS could have a higher density in sulfate groups. Thus, we estimate that the order of increasing sulfation of the GAGs used for IFT experiments is: HS < dsLMWH < LMWH.

No significant difference in affinity for HS was detected between the long (KKKQQK) and short (KKQK) GAG-binding motifs (Figure 2). Trimer peptides (KKAK-, KKQKASAS- and KKQK-3mers) showed higher affinity for HS in comparison with dimer peptides (KKAK-, KKQKASAS-, KKQK- and KKKQQK-2mers), indicating the contribution of the additional GAG-binding motif to the HS-binding affinity. This is consistent with the report that peptides containing multiple copies of the GAG-binding motifs XBBXBX and XBBBXXBX had higher affinity for heparin [30]. It is important to point out that CXCL9(74-103) had the highest affinity for HS, whereas it had only two GAG-binding motifs. Hence, not only the number, but also the type of GAG-binding motifs seems to determine the affinity for GAGs, as CXCL9(74-103) contains two different GAG-binding motifs, unlike the other peptides that were synthesized. In contrast to the affinities for HS, we observed higher affinities of the KKKQQK-2mer for binding to the two highest sulfated GAGs: LMWH and dsLMWH, in comparison with the KKQK-2mer (Figure 4). Consequently, the BBBXXB GAG-binding motif appears to be an important element for binding to these GAGs. The additional parameters that contribute to the GAG-specificity of peptides include non-basic amino acids in the GAG-binding motifs and linker sequences (Figure 3 and Figure 5). To be specific, Gln residues increased the binding affinity of trimer peptides for LMWH, dsLMWH and HS, whereas Ala residues increased the binding affinity for HA. The importance of the non-basic amino acid residues for high affinity binding to GAGs was also demonstrated for mouse serglycin-derived peptides [30]. A fragment (YPARRARYQWVRCKP) from the native mouse serglycin core protein that contained a BBXB GAG-binding motif had significant heparin-binding affinity. A comparison of this native peptide with a modified peptide (AAARRARAAAARAKA), in which Ala residues replaced all non-basic amino acids, revealed a 350-fold decrease in heparin-binding affinity [30]. As expected, the presence of sulfate groups on the GAGs also appears to be important for the binding of peptides to GAGs, since a lower affinity for dsLMWH in comparison with LMWH was observed for all peptides (Figure 5). Surprisingly, the KKAK-3mer, and not CXCL9(74-103), showed the highest affinity for the non-sulfated HA.

In order to study the binding of peptides to cellular GAGs, a cell line that could function as an appropriate negative control was needed. Therefore, we chose to use CHO-K1 cells and pgsB-618 cells, which do not produce GAGs. CHO cells are known to produce HS, CS and DS [31,32,33]. The pgsB-618 cell line was developed by Esko et al., who treated CHO-K1 cells with ethyl methanesulfonate, and subsequently screened for mutants that were defective in proteoglycan synthesis [22]. Five CHO cell mutants, showing less than 2% of the β4Gal-T7 activity that was observed in wildtype CHO-K1 cells, were isolated. The synthesis of both HS and CS was abolished in these β4Gal-T7-deficient cells [22]. The enzyme β4Gal-T7 is responsible for the attachment of the first galactose in the biosynthesis of the tetrasaccharide linkage region (GlcAβ1–3Galβ1–3Galβ1– 4Xylβ1-O-Ser) that is common to HS, heparin, CS and DS [23,34]. Surprisingly, the pgsB-618 cells appeared to be a mixed population of cells as both the HS^+^ and HS^−^ subpopulations were detected using flow cytometry (Appendix A). Therefore, either this cell line was not derived from a single cell, the cell line had been contaminated with an HS^+^ CHO cell line or a post-cloning genetic event had occurred, resulting in the mixed population of cells [35]. Esko et al. mentioned the possible presence of revertant cells, and therefore only subcultured the cells less than 15 times. However, the cells we received had a low passage number (#8), so the presence of revertant cells at the time the expression of HS was tested initially was very unlikely. Furthermore, Esko et al. reported the β4Gal-T7 activity of the pgsB-618 cells to be 1.6% in comparison with the wildtype CHO cells. After subcloning the pgsB-618 cells, several HS^−^ and HS^+^ clones were obtained. The binding of TAMRA-labeled peptides (KKAK-2mer, KKQK-3mer, KKKQQK-2mer and CXCL9(74-103)) to cellular GAGs was assessed using HS^+^ and HS^−^ CHO cells in parallel. The four peptides bound significantly better to HS^+^ than to HS^−^ CHO cells (Figure 7). A trend towards an increased binding capacity of the KKKQQK-2mer peptide in comparison with the KKQK-3mer for cellular GAGs could be noticed. This observation is not in accordance with the presence of less GAG-binding motifs in the KKKQQK-2mer, suggesting a possible contribution of the BBBXXB GAG-binding motif to the binding capacity for cellular GAGs. Furthermore, the KKAK-2mer and CXCL9(74-103) showed the lowest and highest binding capacity for cellular GAGs, respectively, which is in line with the IFT data. Remarkably, the binding of these four peptides to HS^−^ CHO cells was not entirely abrogated (Figure 7B). The observed residual binding can be described as non-HS-mediated binding and may be explained by the presence of other GAGs or negatively charged molecules on the cell surface, as only the absence of HS was confirmed by flow cytometry. Interestingly, Rueda et al. observed the residual binding of CXCL12γ to both HS-deficient pgsD-677 cells and GAG-deficient pgsA-745 cells [36]. This may be due to the binding of the highly positively charged COOH-terminal domain of CXCL12γ to negatively charged structures such as sulfated glycosphingolipids (sulfatides) or sialylated O-glycans that are present on the cell surface and have been identified as binding partners for chemokines [36,37,38].

The addition of soluble heparin or HS reduced the binding of CXCL9(74-103) to cellular GAGs (Appendix A). The more heparin or HS was added, the less CXCL9(74-103) was bound. This effect could be observed on the HS^+^ CHO cells and, to a lesser extent, on the HS^−^ CHO cells. The effect of heparin was more pronounced than the effect of HS, as lower doses of heparin (0.3, 1 and 3 μg) already resulted in greater reductions in the binding of the peptide to the cellular GAGs. In contrast, the addition of HA did not affect the binding of CXCL9(74-103) to cellular GAGs (Appendix A). As opposed to heparin, HS contains less and HA lacks sulfation. This reduced presence of negatively charged sulfates may explain the less pronounced effect of HS or the failure of HA to compete for peptide binding to cells. These experiments with soluble GAGs emphasize the strong GAG-binding capacity of CXCL9(74-103) on heparin and HS, as well as its GAG-specificity, since it poorly interacted with HA.

The peptides in this study were fluorescently labeled with TAMRA in order to enable their detection in both IFT and flow cytometry experiments. Molecular interactions may be affected by any kind of labeling. However, all the peptides were labeled with the same fluorescent group at the same position (N-terminus), allowing a direct comparison between the different peptides.

In summary, this study provides insights into the structural elements, such as the number and type of GAG-binding motifs and linker sequences, in peptides that play a role in GAG-binding affinity. Moreover, we report amino acid motifs that give specificity for binding to particular GAGs. An increased understanding of GAG-binding motifs will facilitate their identification in GAG-binding proteins, leading to increased insights into GAG–protein interactions, which is important for therapeutic drug development targeting these interactions.

## 4. Materials and Methods

### 4.1. Solid-Phase Synthesis of Peptides

Eight different peptides (Table 1), namely KKAK-2mer and -3mer, KKQKASAS-2mer and -3mer, KKQK-2mer and -3mer, KKKQQK-2mer and CXCL9(74-103), were chemically synthesized based on 9-fluorenyl methoxycarbonyl (Fmoc) chemistry, as described by Loos et al. [39]. Peptide synthesis was initiated on an Fmoc-Thr(But)-Wang resin (100–200 mesh, 1% DVB; Activotec; Cambridge, UK) using an Activo-P11 automated peptide synthesizer (Activotec; Cambridge, UK). For interaction studies, the peptides were site-specifically labeled with the fluorescent dye TAMRA (Novabiochem, Merck; Darmstadt, Germany) at the NH_2_-terminus before the removal of the amino acid side-chain protection groups.

The peptides were purified by reversed-phase high-performance liquid chromatography (RP-HPLC) using a 250 × 8 mm Pepmap C18 5 μm column (VDS optilab; Berlin, Germany) and a 150 × 10 mm Proto 300 C18 5 μm column (Higgins Analytical; Mountain View, CA, USA). After elution in an acetonitrile gradient in 0.1% (*v*/*v*) trifluoroacetic acid or 0.1% (*v*/*v*) acetic acid, the peptides were detected using on-line ion trap mass spectrometry (AmaZon SL; Bruker; Bremen, Germany). The purity of the individual HPLC fractions of the peptides was assessed using ion trap mass spectrometry (on AmaZon SL or AmaZon Speed ETD instruments; Bruker; Bremen, Germany). Deconvolution software (Bruker; Bremen, Germany) was used to experimentally determine the relative molecular mass (M_r_) of the peptides. For each peptide, pure fractions were pooled, lyophilized and dissolved in ultrapure water (Arium pro; Sartorius; Goettingen, Germany).

### 4.2. Production of Desulfated LMWH

The selective desulfation of 2-O- and 3-O-sulfates of LMWH was performed according to the protocol of Fryer et al. [40]. First, 5 mg of enoxaparin was dissolved in 30 μL of deionized water. Next, 10 μL of 2 M NaOH was added to reach a final NaOH concentration of 0.5 M, and the solution was mixed, frozen in liquid nitrogen and lyophilized overnight. Subsequently, the solution was neutralized with 2 M acetic acid and buffered to pH 7.2 by the addition of 450 μL of phosphate buffered saline (PBS; pH 7.2; 10 mM phosphate buffer and 137 mM NaCl). The solution was desalted for 1.5 days in a 0.5 mL float-a-lyzer (500 Da molecular weight cut-off) against 500 mL of deionized water (the water was exchanged two times). The desalted solution was frozen in liquid nitrogen and lyophilized for 2 days. The dried GAG was weighed out and dissolved in deionized water. The degree of desulfation was assessed by measuring the heparin activity using an anti-factor Xa (Kinetichrome) assay (Iduron; Cheshire, UK). Heparin’s anti-Xa activity was determined enzymatically using a chromogenic peptide substrate that produces a chromophore (405 nm) whose wavelength is inversely related to the activity of heparin in the test sample. Using the desulfation protocol, a 77.8% reduced activity of dsLMWH was observed compared with commercial LMWH. Additionally, strong anion exchange chromatography was performed to compare the disaccharide composition, as described in a recent study [41]. In short, dsLMWH and LMWH were digested using heparinase III and the samples were subjected to strong anion exchange chromatography. By comparing the retention times to commercially available disaccharide standards (Iduron) with different degrees of sulfation, the disaccharide composition could be determined, and the peak areas were compared relatively between dsLMWH and its control. It should be noted that no 3-O desulfated disaccharide standard exists, thus only 2-O-desulfation can be monitored here. The changes in relative molar disaccharide composition depicted a 45% reduction in 2-O-sulfations, with the UA2S-GlcNS6S (38% reduction) disaccharide being the most affected.

### 4.3. Isothermal Fluorescence Titration (IFT)

In order to study the affinity of the interaction between chemically synthesized TAMRA-labeled peptides and GAGs, isothermal fluorescence titration (IFT) experiments were performed [42,43,44,45,46]. Upon the interaction of a peptide with a GAG, a structural re-arrangement of the peptide is induced, influencing the fluorescence emission. Consequently, a dose-dependent decrease in fluorescence (quenching) could be detected. Hence, the change in fluorescence intensity was used to quantify the binding of the peptide to the GAG. The K_d_ values were calculated from the resulting binding isotherms.

A spectrofluorometer (Fluoromax-4; Horiba Scientific; Kyoto, Japan), attached to a temperature controller (LFI-3751; Wavelength Electronics; Bozeman, MT, USA) to ensure a stable temperature of 20 °C, was used for the measurements. Fluorescence emission spectra were recorded at 560 to 750 nm after excitation at 550 nm. The scan speed was set to 500 nm/min, and the slit widths were set at 3 nm and 5 nm for excitation and emission, respectively. 

Concentrated stock solutions (100 μM and 500 μM) of GAGs were prepared, and the GAGs used in these experiments included: LMWH (5 kDa; Lovenox; Sanofi; Bridgewater, NJ, USA), HS (22 kDa; Celsus Laboratories; Cincinnati, OH, USA), HA (11.5 kDa; Sigma-Aldrich; St. Louis, MO, USA) and partially desulfated LMWH (5 kDa; dsLMWH; 2-O, 3-O desulfated enoxaparin; desulfated in the laboratory in Graz).

First, a fluorescence emission spectrum of the TAMRA-labeled peptide at 700 nM in 500 μL PBS without any added GAG was recorded after equilibration at room temperature in the dark for 30 min (intrinsic fluorescence of the peptide). Next, 0.5 μL of a 100 μM GAG stock solution was added, and the next spectrum was recorded after 1 min of equilibration. This step was repeated until a total GAG concentration of 800 nM was reached. Subsequently, the highest concentration (500 μM) of GAG stock solution was added in 0.5 μL steps until a total GAG concentration of 3300 nM was obtained. These titration series were performed three times in total. For each individual measurement, the area under the curve (AUC) was calculated from the fluorescence emission spectrum. To eliminate background signals, fluorescence emission spectra of the titration series without peptide (in PBS only) were also collected, and the corresponding AUC values were subtracted from those of the spectra with peptide [45].

The change in fluorescence intensity (ΔF) divided by the intrinsic fluorescence of the peptide (F_0_ corresponding to the fluorescence emission in the absence of GAG) was plotted against the concentration of added GAG. ΔF is acquired by subtracting F_0_ from the fluorescence emission measured at a certain GAG concentration (F). Mean values of ΔF/F_0_ were calculated from three independent measurements. Origin 8.0 software (OriginLab; Northampton, MA, USA) was used to analyze the resulting binding isotherms. Data fitting was performed by applying a nonlinear regression using the following equation:$$F=F_{i}+F_{max}\left[\frac{K_{d}\;+\;\left[protein\right]\;+\;\left[ligand\right]\;-\;\sqrt{{\left(K_{d}\;+\;\left[protein\right]\;+\;\left[ligand\right]\right)}^{2}\;-\;4\left[protein\right]\left[ligand\right]}}{2\left[protein\right]}\right]$$
where F_i_ and F_max_ are the values of the initial fluorescence, which was fixed to 0 for the calculation of the fitting and the maximum fluorescence, respectively; and [protein] and [ligand] represent the total protein (TAMRA-labeled peptide) and ligand (GAG) concentrations [45]. The dissociation constants (K_d_ values) were calculated and used for data comparison and are presented as values +/− standard deviation (SD), where SD refers to the precision of the fitting procedure. This fitting equation assumes an equimolar ratio of binding sites on the peptide and its ligand. However, since there is a mismatch of binding sites between peptides and GAGs, due to the molecular size difference, this was taken into account by correcting the protein concentration according to the number of potential GAG-binding motifs. 

### 4.4. Cells

Chinese hamster ovary (CHO)-K1 cells (ATCC CCL-61) and pgsB-618 CHO cells (ATCC CRL-2241; provided by Professor Gertrud Malene Hjortø) were used. CHO cell lines were cultured in Ham’s F-12 Nutrient Mix (Gibco, ThermoFisher Scientific; Waltham, MA, USA) supplemented with L-glutamine, 10% (*v*/*v*) fetal calf serum (FCS), 1 mM sodium pyruvate (Gibco, ThermoFisher Scientific; Waltham, MA, USA) and 0.1% (*v*/*v*) sodium bicarbonate (Gibco, ThermoFisher Scientific; Waltham, MA, USA) in 5% CO_2_ at 37 °C.

### 4.5. Cellular Surface HS Expression

The expression of HS on the cell surface of CHO-K1 and pgsB-618 cells was measured using flow cytometry. First, the medium was discarded, the cells were washed with PBS (BioWhittaker DPBS without calcium and magnesium; Lonza; Verviers, Belgium) and were detached with trypsin/EDTA at 37 °C. Subsequently, the cells were resuspended in culture medium and left at room temperature for 1 h. After centrifugation at 177 g and 4 °C for 7 min, the pellet was resuspended in PBS and the cells were kept on ice. For each condition, 3 × 10^5^ cells in 100 μL PBS were used and 4.7% (*v*/*v*) FcR blocking reagent (human; Miltenyi Biotec; Bergisch Gladbach, Germany) and 0.06% (*v*/*v*) Zombie Aqua fixable viability dye (Biolegend; San Diego, CA, USA) were added to the cells. Following incubation in the dark at room temperature for 20 min, a wash step was performed: the cells were washed with flow cytometry buffer (PBS supplemented with 2% (*v*/*v*) FCS and 2 mM EDTA; 1 mL per tube), centrifuged at 299 g and 4 °C for 5 min, and the supernatants were discarded. Afterwards, the cells were incubated with mouse anti-HS antibody (clone F58-10E4; Amsbio; Abingdon, UK) in the dark on ice for 30 min. After another wash step with flow cytometry buffer (vide supra), the cells were incubated with a secondary R-phycoerythrin goat anti-mouse IgM antibody (Jackson ImmunoResearch; Ely, UK) in the dark on ice for 30 min. Finally, a last wash step with flow cytometry buffer (vide supra) was performed prior to the fixation of the cells using the flow cytometry buffer supplemented with 0.4% (*v*/*v*) formaldehyde. The cells were stored at 4 °C until flow cytometric analysis was performed using a BD LSRFortessa X-20 flow cytometer (BD Biosciences; Franklin Lakes, NJ, USA). The results were analyzed using FlowJo software (BD Biosciences; Franklin Lakes, NJ, USA).

### 4.6. Subcloning of pgsB-618 Cells

PgsB-618 CHO cells were diluted and subsequently seeded in 96-well plates at densities of 5, 1 or 0.5 cells/well in 150 μL medium per well. The wells were screened for the presence of single colonies of cells using light microscopy. A total of 21 wells that each contained a single colony of cells were selected, and the cells were transferred to the wells of a 24-well plate, after detaching them with 0.05% trypsin/EDTA (Gibco, ThermoFisher Scientific; Waltham, MA, USA). Upon reaching confluence, the cells were transferred to the wells of 6-well plates and subsequently to T25 cell culture flasks to enable the cell expansion of the 21 subclones. The HS expression on the cell surface of each of the 21 subclones was measured using flow cytometry.

### 4.7. Binding of Peptides to Cellular GAGs

The binding of four different TAMRA-labeled peptides (KKAK-2mer, KKQK-3mer, KKKQQK-2mer and CXCL9(74-103)) to cellular GAGs on pgsB-618 clone 16 (HS^−^ CHO cells) and pgsB-618 clone 19 cells (HS^+^ CHO cells) was assessed using flow cytometry. Since each peptide contains exactly one TAMRA-group connected to the N-terminus, this allows us to quantify the peptides by their fluorescence or absorbance.

First, the medium was discarded, the cells were washed with PBS and were detached with trypsin/EDTA at 37 °C. Subsequently, the cells were resuspended in medium and left at room temperature for 45 min, and afterwards on ice for 15 min. After centrifugation at 177 g and 4 °C for 7 min, the pellet was resuspended in PBS and the cells were kept on ice. For each condition, 1 × 10^5^ cells in 100 μL PBS were used and 4.7% (*v*/*v*) FcR blocking reagent and 0.06% (*v*/*v*) Zombie Aqua fixable viability dye were added to the cells. Following incubation in the dark at room temperature for 20 min, a wash step with flow cytometry buffer was performed. Afterwards, 0.24 nmol TAMRA-labeled peptide was added to the cells. After incubation in the dark on ice for 30 min, a wash step with flow cytometry buffer was performed and, subsequently, the cells were fixed with 0.4% (*v*/*v*) formaldehyde in flow cytometry buffer and kept at 4 °C. Flow cytometric analysis was performed within the following 24 h using a BD LSRFortessa X-20 flow cytometer (BD Biosciences; Franklin Lakes, NJ, USA), and the results were analyzed using FlowJo software (BD Biosciences).

Similarly, the binding of three TAMRA-labeled peptides (KKQK-3mer, KKKQQK-2mer and CXCL9(74-103)) to cellular GAGs on pgsB-618 clone 16 (HS^−^ CHO cells) and pgsB-618 clone 19 cells (HS^+^ CHO cells) in the presence of added GAGs was assessed using flow cytometry. In this experiment, different doses (0.03, 0.1, 0.3, 1 or 3 μg) of heparin (heparin low in calcium; Iduron; Cheshire, UK), HS (GAG HS01 BN1; Iduron; Cheshire, UK) or HA (70–120 kDa; HA sodium salt 96144; Sigma-Aldrich; St. Louis, MO, USA) were added to the cells, simultaneously with 0.24 nmol of TAMRA-labeled peptide (vide supra).

Flow cytometry data (Figure 7C and Appendix A) are represented as median. Statistical analysis of these data consisted of a Shapiro–Wilk test to assess the normality of the different groups. As some groups did not pass the normality test, a nonparametric Kruskal–Wallis test was subsequently performed to evaluate if the median values of the groups varied significantly. In the next step, either a Mann–Whitney test (for unpaired data) or a Wilcoxon test (for paired data) was used to compare the groups. Statistical analysis was performed using GraphPad Prism 8.3.0 software (GraphPad Software; San Diego, CA, USA).

## Figures and Tables

**Figure 1 ijms-23-00447-f001:**
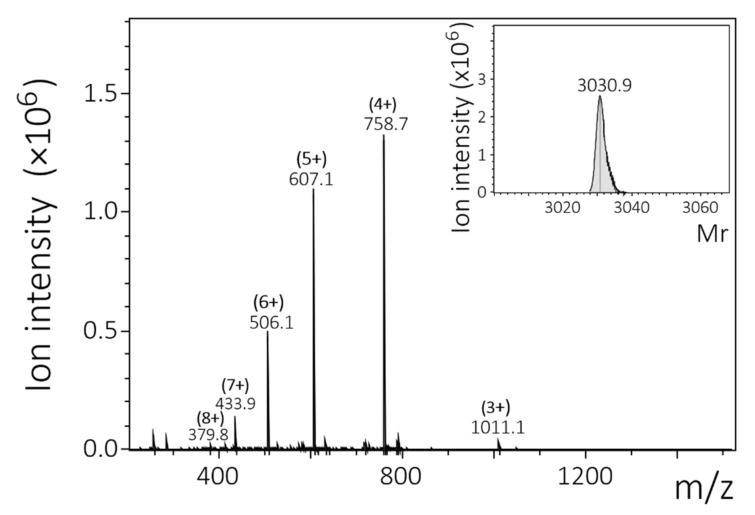
Identification of the chemically synthesized TAMRA-KKQK-3mer. The peptide was synthesized using Fmoc chemistry. After deprotection, RP-HPLC was used to purify the peptide and detection was performed using ion trap mass spectrometry. A mass spectrum of the peptide is shown with the intensity of the detected ions on the y-axis and the mass/charge (*m*/*z*) ratios on the x-axis. Deconvolution software (Bruker) was used to determine the relative molecular mass (M_r_) of the uncharged peptide (insert).

**Figure 2 ijms-23-00447-f002:**
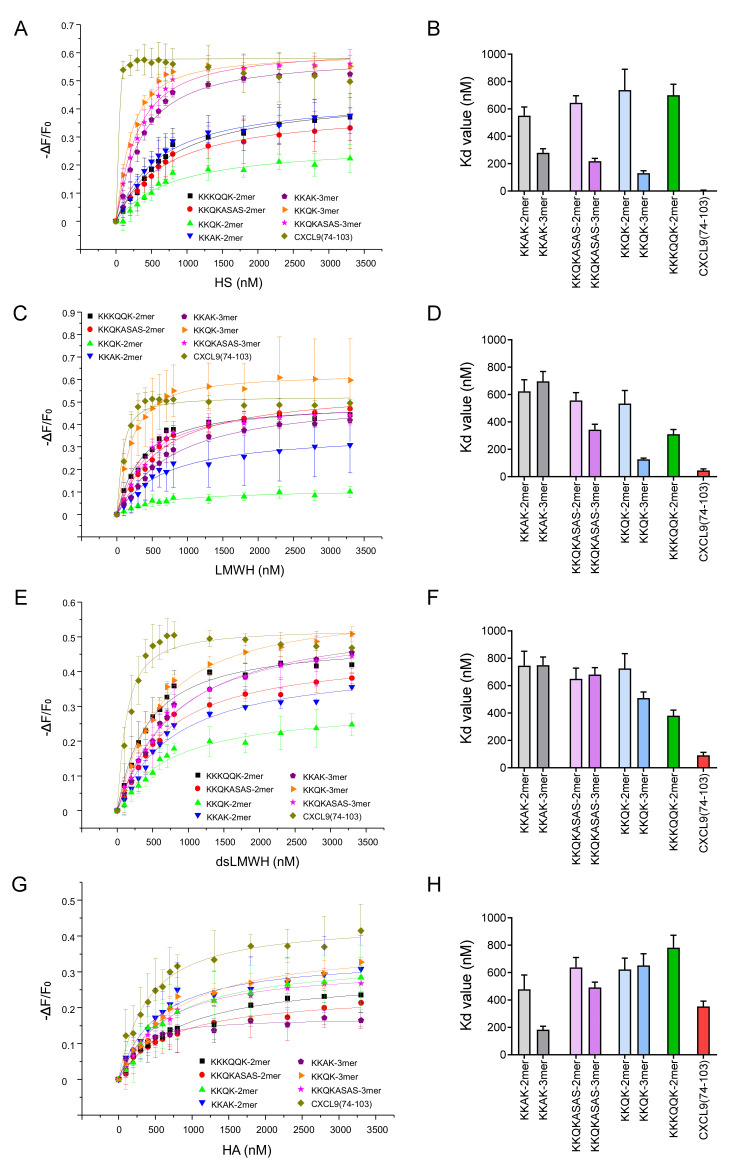
Peptides with more GAG-binding motifs have higher affinity for HS. Isothermal fluorescence titration (IFT) measurements were performed to study the interactions between TAMRA-labeled peptides and different GAGs. (**A**) Binding isotherms for the different peptides are displayed per GAG: (**A**) HS; (**C**) LMWH; (**E**) dsLMWH; (**G**) HA. The change in fluorescence intensity (ΔF) divided by the intrinsic fluorescence of the peptide (F_0_) is shown on the y-axis. ΔF is calculated by subtracting F_0_ from the fluorescence emission measured at a certain GAG concentration (**F**). The concentration of added GAG is shown on the x-axis. Data points represent mean values ± SEM (*n* = 3). Dissociation constant (K_d_) values (±SD) that are derived from the binding isotherms are visualized in bar plots per GAG: (**B**) HS; (**D**) LMWH; (**F**) dsLMWH; (**H**) HA.

**Figure 3 ijms-23-00447-f003:**
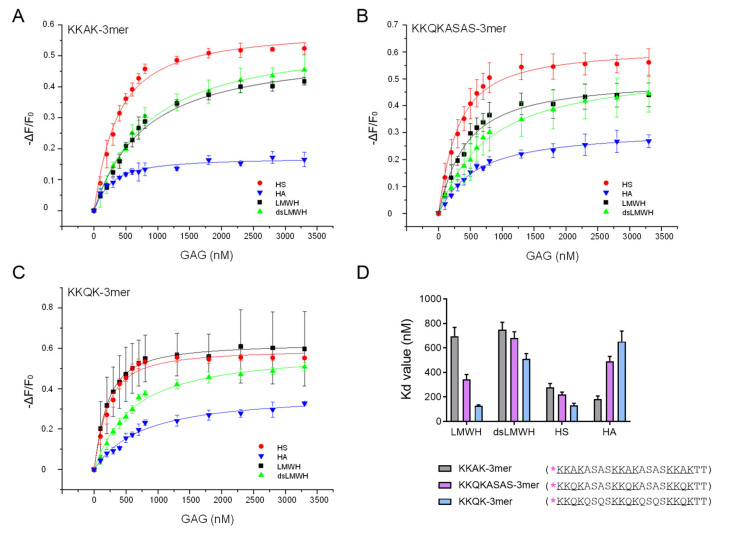
Gln residues in both GAG-binding motifs and linker sequences increase trimer peptide binding affinity for (ds)LMWH and HS, but not HA. Isothermal fluorescence titration (IFT) measurements were performed to study the interactions between TAMRA-labeled trimer peptides (KKAK-, KKQKASAS- and KKQK-3mer) and GAGs ((ds)LMWH, HS and HA). (**A**–**C**) Binding isotherms for the KKAK-3mer (**A**), KKQKASAS-3mer (**B**) and KKQK-3mer (**C**) are displayed. The change in fluorescence intensity (ΔF) divided by the intrinsic fluorescence of the peptide (F_0_) is shown on the y-axis. ΔF is calculated by subtracting F_0_ from the fluorescence emission measured at a certain GAG concentration (F). The concentration of the added GAG is shown on the x-axis. Data points represent mean ± SEM (*n* = 3). (**D**) K_d_ values (±SD) that are derived from the binding isotherms in (**A**–**C**) are visualized in the bar plot. TAMRA-labeling is indicated with [*] and the GAG-binding motifs are underlined in the amino acid sequences of the peptides.

**Figure 4 ijms-23-00447-f004:**
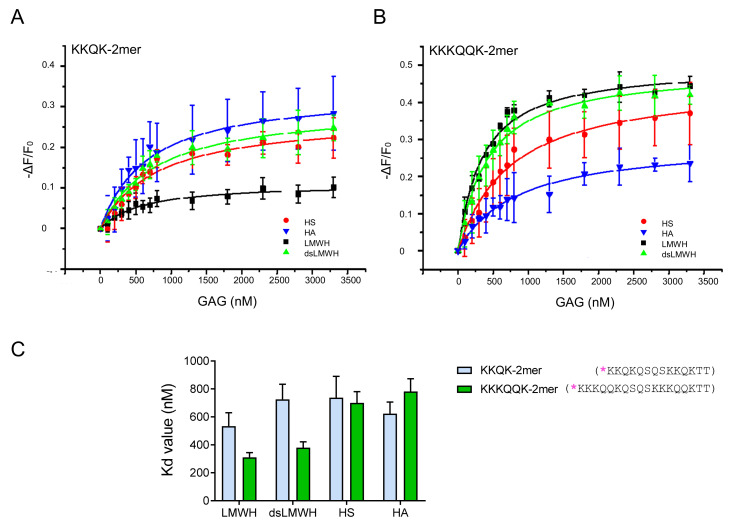
Dimer peptides with either short or long GAG-binding motifs show differential affinity for GAGs. Isothermal fluorescence titration (IFT) measurements were performed to study the interactions between TAMRA-labeled dimer peptides (KKQK-2mer and KKKQQK-2mer) and GAGs ((ds)LMWH, HS and HA). (**A**,**B**) Binding isotherms for the KKQK-2mer (**A**) and the KKKQQK-2mer (**B**) are displayed. The change in fluorescence intensity (ΔF) divided by the intrinsic fluorescence of the peptide (F_0_) is shown on the y-axis. ΔF is calculated by subtracting F_0_ from the fluorescence emission measured at a certain GAG concentration (F). The concentration of the added GAG is shown on the x-axis. Data points represent mean ± SEM (*n* = 3). (**C**) K_d_ values (±SD) that are derived from the binding isotherms in (**A**,**B**) are visualized in the bar plot. TAMRA-labeling is indicated with [*] and the GAG-binding motifs are underlined in the amino acid sequences of the peptides.

**Figure 5 ijms-23-00447-f005:**
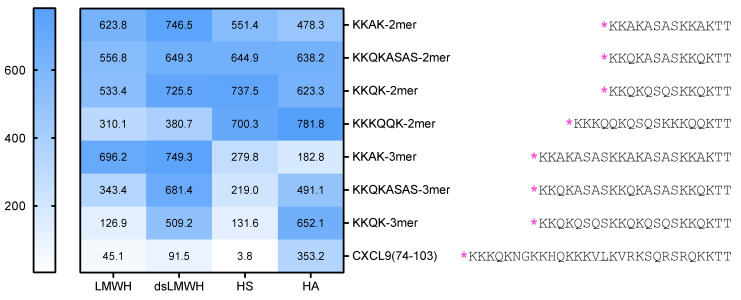
Overview of the K_d_ values obtained from the IFT experiments. The K_d_ values of the interactions of the different peptides with (ds)LMWH, HS and HA are displayed in nM. The amino acid sequences of the peptides are depicted on the right with TAMRA indicated in pink [*] and the GAG-binding motifs underlined.

**Figure 6 ijms-23-00447-f006:**
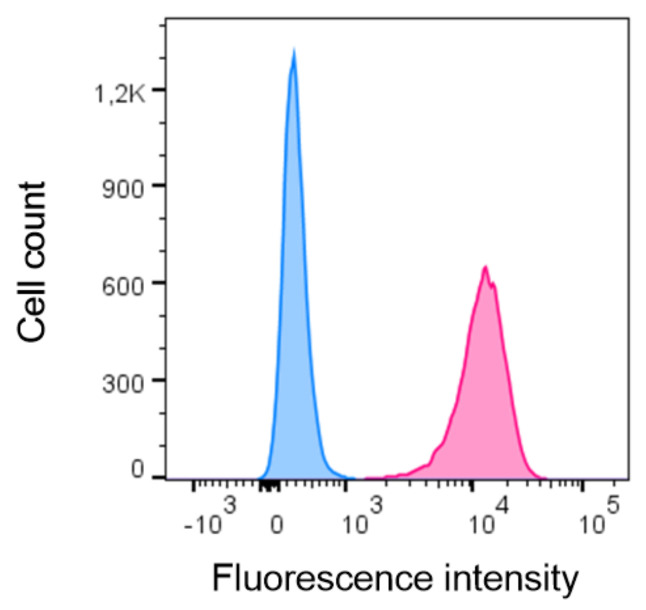
Subcloning of pgsB-618 cells results in HS^+^ (pgsB-618 clone 19) and HS^−^ (pgsB-618 clone 16) cells. The expression of HS on pgsB-618 clone 16 (blue) and pgsB-618 clone 19 (pink) cells was measured using flow cytometry with a primary anti-HS antibody and a secondary PE-labeled antibody. A histogram is shown with the cell count on the y-axis and the fluorescence intensity of the secondary antibody on the x-axis.

**Figure 7 ijms-23-00447-f007:**
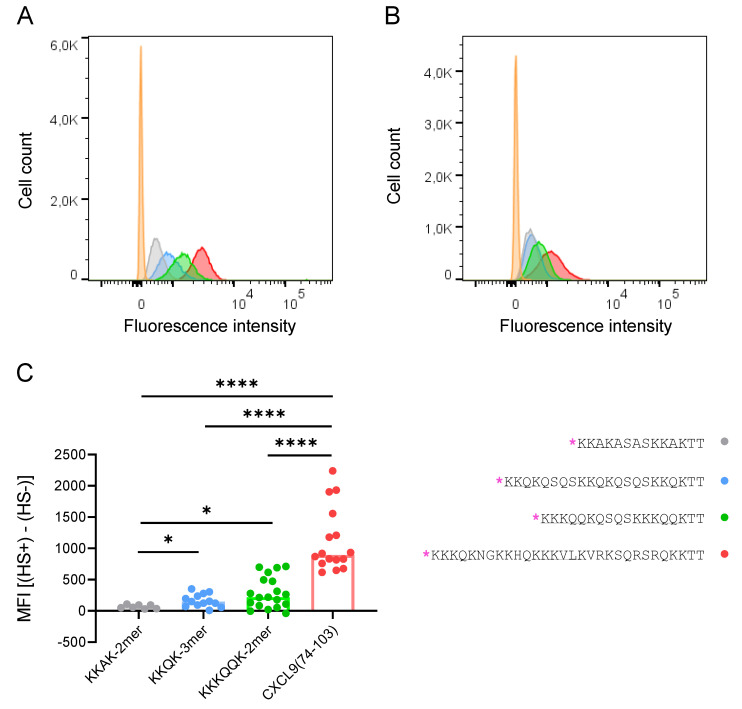
Peptides show differential binding capacity for cellular GAGs on CHO cells. TAMRA-labeled peptide (0.24 nmol; KKAK-2mer (gray), KKQK-3mer (blue), KKKQQK-2mer (green), CXCL9(74-103) (red), unstained (orange)) was added to either HS^+^ or HS^−^ CHO cells and binding was assessed using flow cytometry. (**A**,**B**) Histograms (of one representative experiment) of the HS^+^ (**A**) and HS^−^ (**B**) CHO cells are shown with the cell count on the y-axis and the fluorescence intensity of the TAMRA-labeled peptides on the x-axis. (**C**) Median fluorescence intensity (MFI) values were normalized first by subtracting the MFI of the unstained sample (background). The difference between the MFI of the HS^+^ and the HS^−^ CHO cells is displayed (y-axis) for each peptide (x-axis). (*n* ≥ 7; Mann–Whitney test, * *p* < 0.05, **** *p* < 0.0001 (comparison of the different peptides)). The amino acid sequences of the peptides are depicted on the right with TAMRA indicated in pink [*] and the GAG-binding motifs underlined.

**Table 1 ijms-23-00447-t001:** Overview of the chemically synthesized peptides.

Peptide Name	Amino Acid Sequence	Theoretical Average M_r_	Experimental M_r_
TAMRA ^1^-KKAK-2mer	* KKAKASASKKAKTT	1859.2	1859.7
TAMRA-KKAK-3mer	* KKAKASASKKAKASASKKAKTT	2631.1	2631.8
TAMRA-KKQK-2mer	* KKQKQSQSKKQKTT	2087.4	2087.3
TAMRA-KKQK-3mer	* KKQKQSQSKKQKQSQSKKQKTT	3030.4	3030.9
TAMRA-KKQKASAS-2mer	* KKQKASASKKQKTT	1973.3	1973.1
TAMRA-KKQKASAS-3mer	* KKQKASASKKQKASASKKQKTT	2802.2	2802.4
TAMRA-KKKQQK-2mer	* KKKQQKQSQSKKKQQKTT	2600.0	2599.3
TAMRA-CXCL9(74-103)	* KKKQKNGKKHQKKKVLKVRKSQRSRQKKTT	4072.8	4072.9

^1^ The produced peptides were labeled with 5(6)-carboxytetramethylrhodamine (TAMRA) at the NH_2_-terminus (indicated in pink [*]). Glycosaminoglycan (GAG)-binding motifs are underlined.

## Data Availability

All data are included in the text and Appendix A of the manuscript.

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
