# Peer review of "Affinity and Specificity for Binding to Glycosaminoglycans Can Be Tuned by Adapting Peptide Length and Sequence"

_ijms, 2021, doi:10.3390/ijms23010447_

Round 1

Reviewer 1 Report

The paper by Crijns H. and coworkers deals with the study of the structural features of basic peptides that contribute to their heparin-binding capacity.

Although the topic of this paper has been the subject of plenty of published articles in the past, this manuscript contains enough interesting new findings to be considered for publication in IJMS, provided the modifications/implementations requested here below.

1) 2, lane 83. I’m not sure that the proximity of basic amino acids depends (exclusively) on the secondary structure (alpha helices and beta sheets) of the protein. Also the ternary structure (folding driven by S-S bridges) contributes to the formation of the heparin-binding domains.

The importance of protein folding in generating heparin binding domains must be introduced and discussed referring to the appropriate examples of proteins containing linear or conformational basic domains.

Experimentally, the influence of a proper tridimensional structure of those peptides containing enough cysteine residues to allow a (possible) conformational folding must be assessed (i.e. assaying their heparin-binding capacity in the experimental conditions adopted before or after thermal denaturation or in the presence of S-S disrupting agents.

2) It is not clear why the authors decided to use low molecular weight heparin instead of standard heparin for their experiments.

Also, no structural features of the GAG used are reported: the average length of the GAGs must be provided, along with the degree of desulfation obtained by the in-house desulfation procedure. I strongly suggest the authors to provide these data in a dedicated paragraph in the material and methods section.

These structural features are mandatory to fully interpreter the results and straighten authors’ conclusions. As examples: The proposed comparison between the results obtained with LMWH and HS (pg. 13, lane 391-392) cannot be done with an “in general” introduction, but must take in account the real degree of sulfation of the two reagents used in this work and, equally important, their length.

Similarly, I don’t find appropriate the sentence on pg 13, lanes 392-395, where a conclusion is drawn on a degree of sulfation that is only “expected”.

3) The meaning of the results from the experiments described in paragraph 2.7 is not clear. This paragraph completely lacks any meaningful interpretation that allows the reader to connect these experiments (and related results) to the rest of the work. In discussion (pg. 14, lanes 473-474), the conclusions drawn by the authors only confirm data already present in the literature, adding nothing really important to the core issue of the work that is instead focused on the structural features of the protein that contribute to heparin-binding. I think these part of the work can be omitted.

4) Can the authors exclude an interference of the (rather bulky) carboxytetramethylrhodamine tag in the heparin binding of the peptides?

5) I find more appropriate to move the Figure containing the results of the subcloning procedure in the supplementary material and, conversely, include Figures S1-s3 in the main text.

Minor points

1) I’ve found myself repeatedly forced to come and go to figure 1 to correlate the names of the peptides to their sequences. Whenever possible, it should be better to directly report the sequences in the figures since it would really help the reader. This is particularly important for Fig. 5, based on its “overview” nature.

2) Pg. 5, lanes 162-166: the description of the physical bases of the IFT technology does not belong here, better move it to material and methods where its brief description might be come along with a couple of appropriate reference for those who want to deepen the study.

3) The meaning and utility of lanes 287-289 of pg. 9 are not clear. If required, please rephrase and move it in material and method section.

4) In the description of the results of the binding experiments on CHO cells I suggest to use the term “binding capacity” instead of “affinity” (a parameter that has not been evaluated indeed).

5) It is not clear which kind of heparin has been used for the experiments in paragraph 2.7.

Author Response

Please see attached PDF file with a point-by-point reply.

Reviewer 2 Report

Is an interesting study with an impact on therapeutic drug development. The introduction of the article is well structured with many data from the literature that are relevant to this study. The results are well structured, the methods used are adequate to elucidate structural requirements for binding of proteins to glycosaminoglycan. I consider it publishable in this form.

Author Response

We thank the reviewer for the positive evaluation of our study.